# Unleashing the Power of Visual Prompting At the Pixel Level

**Junyang Wu**[*]                                                     *SJTUwjy@sjtu.edu.cn*
*Shanghai Jiao Tong University*

**Xianhang Li**[*]                                                   *xianhang710@gmail.com*
*UC Santa Cruz*

**Chen Wei**                                                        *weichen3012@gmail.com*
*Johns Hopkins University*

**Huiyu Wang**                                                 *williamwanghuiyu@gmail.com*
*FAIR, Meta*

**Alan Yuille**                                                          *ayuille1@jhu.edu*
*Johns Hopkins University*

**Yuyin Zhou**                                                     *zhouyuyiner@gmail.com*
*UC Santa Cruz*

**Cihang Xie**                                                          *cixie@ucsc.edu*
*UC Santa Cruz*

**Reviewed on OpenReview:** *https://openreview.net/forum?id=YqitLs4nHa*

## Abstract

This paper presents a simple and effective visual prompting method for adapting pre-trained models to downstream recognition tasks. Our approach is underpinned by two key designs. First, rather than directly adding together the prompt and the image, we treat the prompt as an extra and independent learnable entity. We show that the strategy of reconciling the prompt and the image matters, and find that warping the prompt around a properly shrinked image empirically works the best. Second, we re-introduce two "old tricks" commonly used in building transferable adversarial examples, *i.e.*, input diversity and gradient normalization, into the realm of visual prompting. These techniques improve optimization and enable the prompt to generalize better. We provide extensive experimental results to demonstrate the effectiveness of our method. Using a CLIP model, our prompting method registers a new record of **82.5%** average accuracy across 12 popular classification datasets, substantially surpassing the prior art by **+5.2%**. It is worth noting that such performance not only surpasses linear probing by **+2.2%**, but, in certain datasets, is on par with the results from fully fine-tuning. Additionally, our prompting method shows competitive performance across different data scales and against distribution shifts.

## 1 Introduction

Deep learning models have witnessed pre-training on increasingly large-scale datasets as a general and effective pathway to succeed in both computer vision (He et al., 2022; Radford et al., 2021; Bao et al., 2021) and natural language processing (Devlin et al., 2018; Brown et al., 2020; Liu et al., 2019). These pre-trained models are termed foundation models (Bommasani et al., 2021). While fully fine-tuning stands as one of the most prevalent paradigms to effectively adapt these foundation models to a range of downstream tasks, it can be computationally intensive due to the large number of training parameters. This has led to a need for more efficient alternatives for adapting these cumbersome foundation models to new tasks.

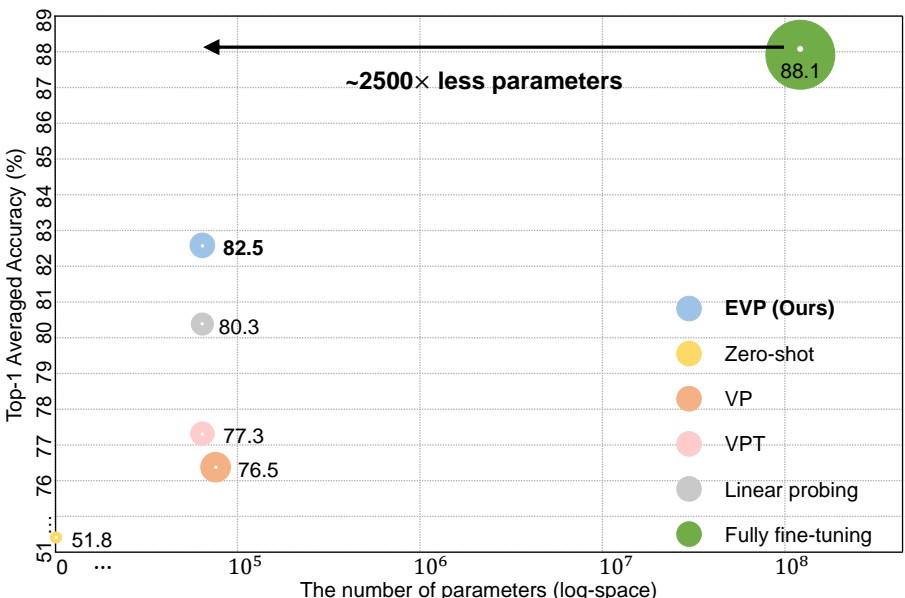

Figure 1: The trade-off between the number of parameters and the average accuracy (across 12 datasets). Our method outperforms linear probing and other visual prompting baselines by a significant margin with a similar number of parameters.

*Prompting* method, which only modifies the input space, offers an effective and efficient solution in NLP (Gao et al., 2021; Lester et al., 2021; Li & Liang, 2021), *e.g.*, text prompting can closely match the performance of fully fine-tuning (Liu et al., 2021b). This promising result has motivated researchers to probe whether similar success can be achieved in the field of computer vision. Some early efforts in this direction include VPT (Jia et al., 2022) and VP (Bahng et al., 2022), which add small amounts of learnable parameters as tokens or perturbations directly at the pixel level to adapt foundation models. However, when taking a closer look at the trade-off between performance and parameter efficiency as shown in Figure 1, we note these advanced visual prompting methods appear to be less competitive. To illustrate, despite having a comparable number of parameters, there exists a significant performance gap between VPT and the simple linear probing baseline (**77.3%** ***vs.*** **80.3%**). In this paper, *we aim to unleash the full potential of visual prompting at the pixel level, and, more importantly, to explore whether it can be stronger than other alternatives such as linear probing.*

Intriguingly, we find that with proper modifications, visual prompting can evolve into a truly effective paradigm for adapting foundation models to different visual tasks. The first key observation is that directly adding together the prompt and the image, as in VP (Bahng et al., 2022), may inadvertently distort the intrinsic information of the image, thereby limiting the prompt's learning potential. We provide a simple strategy to resolve this issue: we first shrink the original image into a smaller size and then pad the prompt around it, *i.e.*, the prompt and the image now are kept as separate entities, free from overlap. This strategy allows for independent optimization of the prompt and enables flexibility in adjusting the padding size to control computation overheads. In addition, we draw on techniques from adversarial examples, which have similarities to visual prompting in their aim to either maximize or minimize the loss function (Bahng et al., 2022; Elsayed et al., 2018), to further improve the performance of visual prompting. Specifically, we find that gradient normalization (Goodfellow et al., 2015; Dong et al., 2018) and input diversity (Xie et al., 2019) are effective at improving the generalization ability of the prompt.

We follow the standard evaluation protocol to conduct experiments across 12 visual benchmarks. We note that, with a CLIP model (Radford et al., 2021), our method attains an averaged accuracy of 82.5%, significantly outperforming the previous state-of-the-art visual prompting method VPT (Jia et al., 2022) by +**5.2**%. More excitingly, this 82.5% result is +**2.2**% stronger than the linear probing result (80.3%) and even comparable to fully fine-tuning on certain datasets. We further confirm the superiority of our method in learning with data at different scales and in handling out-of-distribution samples. We hope our study can catalyze further research in the field of visual prompt learning.

## 2 Related Works

**Prompt learning in NLP.** The key idea of prompting is to reformulate the input text in downstream tasks so that the frozen language models can better *"understand"* and perform the task (Liu et al., 2021a). Prior works (Brown et al., 2020; Petroni et al., 2019; Cui et al., 2021) show that manually designed text prompt can help language models achieve remarkable representation capacity in the few-shot or even zero-shot settings at downstream tasks, but this requires specific domain knowledge. To address this issue, recent works have started to focus on prompt tuning (Li & Liang, 2021; Liu et al., 2021b; Lester et al., 2021), which involves directly optimizing the continuous prompting vector through gradient information. In this work, we investigate prompt learning in computer vision, which is a more challenging task because it involves a different type of signal (visual rather than language) that contains much less high-level semantic information.

**Visual prompt learning.** After witnessing the success of prompting in language models, researchers begin to explore the usage of prompts in the field of computer vision. As a pioneering work, visual prompting is applied as a strategy of model programming Tsai et al. (2020). For example, CoOp (Zhou et al., 2022) applies prompt tuning to vision-language models, learning the soft prompts through minimizing the classification loss on downstream tasks. VP (Bahng et al., 2022) and VPT (Jia et al., 2022) focus on prompting with images: VP optimizes the prompt directly in the pixel space, and VPT proposes to insert a set of learnable tokens into ViT architectures (Dosovitskiy et al., 2020) for prompt tuning. In addition, ILM Chen et al. (2023) and AutoVP Tsao et al. (2024) explore visual prompting from broader perspectives. ILM proposes a label mapping strategy to alleviate the issue of mapping precision. On the other hand, AutoVP considers various aspects such as input scale, visual prompts, pre-trained model selection, and output LM strategies to design high-performance prompt methods automatically. While these approaches show the potential of visual-only prompt learning, as shown in Figure 1, their performance is not as competitive compared to other methods such as linear probing. In this work, we aim to enhance visual prompt learning and demonstrate its strong potential for improving the performance of foundation models on a range of visual tasks.

**Adversarial examples.** It is well-known that machine learning models are vulnerable to adversarial attacks (Dalvi et al., 2004; Biggio et al., 2013; Huang et al., 2011). The fast gradient sign method (FGSM) (Goodfellow et al., 2015) and projected gradient descent (PGD) (Madry et al., 2018) are two commonly used techniques for creating adversarial examples that can fool deep learning models. Nonetheless, these adversarial examples cannot transfer well to fool other models. Later works show that the difficulty of optimizing adversarial examples is the cause of this weak transferability, and techniques such as diverse input patterns (Xie et al., 2019) and momentum-based gradient accumulation (Dong et al., 2017) have been proposed to improve transferability. Given the similarity between generating adversarial examples and the process of prompt learning (Bahng et al., 2022; Elsayed et al., 2018), we hereby are interested in revisiting techniques from building transferable adversarial examples to enhance visual prompting.

## 3 Methodology

In this section, we present **E**nhanced **V**isual **P**rompting (EVP), a simple and effective pixel-level visual prompting method for adapting foundation models to downstream tasks. We first provide a thorough review of previous visual prompting methods as preliminaries, including VP and VPT, and then describe the prompting design and training strategy of our EVP in detail.

### 3.1 Preliminaries

**VPT** (Jia et al., 2022) adds a set of learnable parameters into ViT architecture for visual prompting. For a fair comparison, we hereby consider its VPT-SHALLOW version, which only inserts the prompt into the first layer's input. Specifically, as shown in Figure 2 (a), VPT inserts a collection of prompts $P$ between the learnable class token [CLS] and a sequence of patch embeddings $E$, creating a new input $x = [\text{CLS}, P, E]$.

**VP** (Bahng et al., 2022) adapts foundation models to downstream tasks by directly adding together the learnable prompt and the input images at the pixel level. The prompt $v_\phi$ is designed to be input-agnostic

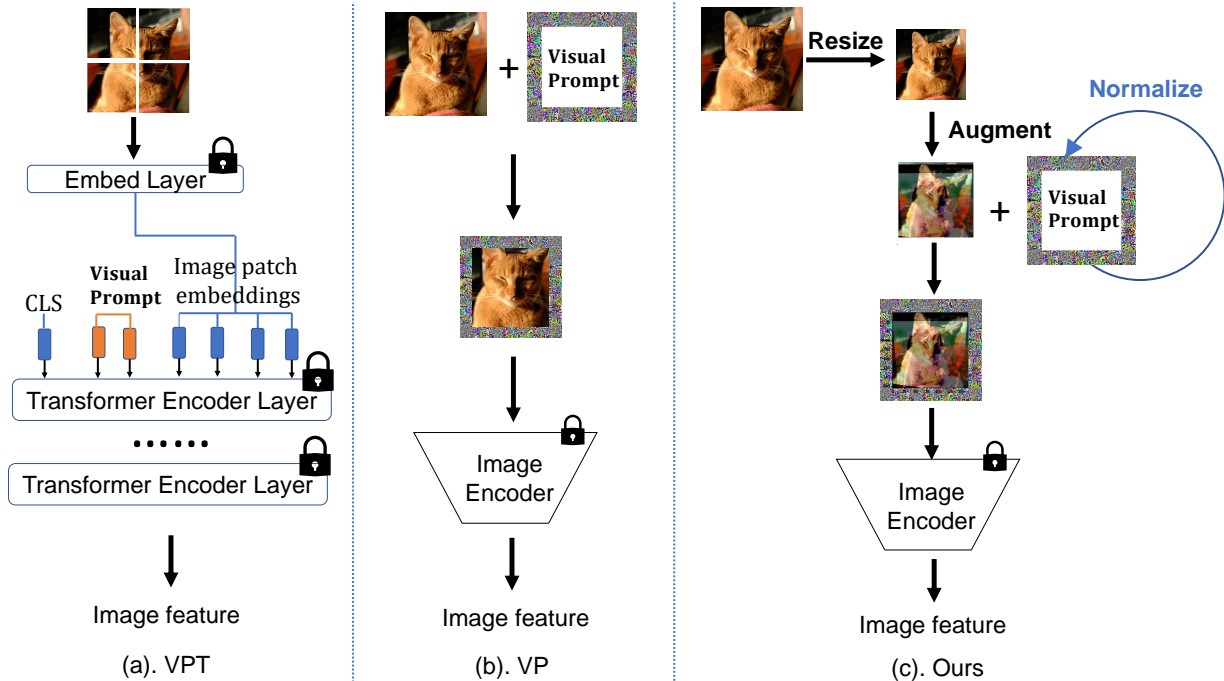

Figure 2: **Overview of the designs of different visual prompting methods.** (a) VPT: Injecting a set of learnable parameters into the token space; (b) VP: Modifying learnable perturbations on the border of input images; (c) Our EVP: Shrinking images and applying data augmentations, then padding the prompt around the image. Note the prompts in EVP are updated using normalization strategies inspired by adversarial attack techniques (Goodfellow et al., 2015).

and task-specific, and is placed on the border of the input images, as shown in Figure 2 (b). During training, VP maximizes the likelihood of the correct label $y$ by optimizing the prompt $v_\phi$: $\max_{v_\phi} P(y|x + v_\phi)$. During inference, the optimized prompt is then added to test images: $X_{test} = \{x_{test}^1 + v_\phi, \dots, x_{test}^n + v_\phi\}$.

## 3.2 Designing EVP

Our prompting design is largely based on VP, but with some simple modifications. The issue we identify with VP is that directly adding together the prompt and the images may corrupt the original image information. For example, in Figure 2 (b), the cat ears are heavily overlapped and obscured by the added prompts. This could hinder the learning of prompts (see our ablations in Section 5.1). To address this issue, as shown in Figure 2 (c), our EVP shrinks the input images and pads the prompt around them. Specifically, for an input image $X \in \mathbb{R}^{K \times K \times 3}$, it is shrunk to $\hat{x} \in \mathbb{R}^{k \times k \times 3}$ and then padded with $(K^2 - k^2) \times 3$ prompts to obtain the output image $\hat{X} \in \mathbb{R}^{K \times K \times 3}$. Similar to VP, during training we optimize the prompt by maximizing the likelihood of the correct label, and during inference we pad the optimized prompt around the shrunk test samples for predictions.

It is important to note that while both EVP and VPT keep the prompt and the image non-overlapping, there is a key difference between these two methods. Specifically, the prompts in EVP are later added with positional embedding, while this is not the case for VPT. As shown in the ablation study in Section 5.2, positional information is crucial for achieving strong performance with visual prompting.

## 3.3 Training Strategy of EVP

There is a strong relationship between prompting and adversarial attacks. In adversarial attacks, the goal is to learn a pixel perturbation $g_i$ that will mislead the network given an image $x_i$. This can be formulated as $\min_{g_i} P(y_i|x_i + g_i)$. On the other hand, visual prompting can be seen as the inverse process of adversarial attacks, in which the aim is to learn a template $v$ that will maximize the likelihood of the correct label $y$.

Given this relationship, we are motivated to explore whether techniques from adversarial attacks, particularly those focused on building transferable adversarial examples, can be useful for visual prompting.

**Input diversity.** Previous work (Xie et al., 2019) demonstrates that input diversity can help optimization and improve the transferability of adversarial examples. As shown in later works, the concept of "diverse input" can be generalized to apply different data augmentation strategies in generating adversarial examples (Dong et al., 2019; Wang et al., 2021; Wu et al., 2021). We hereby re-introduce this concept into visual prompting. Specifically, our EVP considers a range of augmentation including RandomHorizontalFlip, RandomCop, RandAug (Cubuk et al., 2020), and Cutmix (Yun et al., 2019). As shown in Section 5.3, we find that simple augmentation strategies like RandomHorizontalFlip and RandomCrop are already sufficient to significantly improve visual prompting.

**Gradient normalization.** In adversarial attacks, it is common to apply normalization techniques, such as the $L_1$, $L_2$, or $L_\infty$ norm, to the gradients update (Goodfellow et al., 2015). For example, using the $L_2$ norm, the gradient can be normalized as follows:

$$x^{adv} = x + \gamma \frac{\nabla_x J(x, y)}{||\nabla_x J(x, y)||_2} \, , \tag{1}$$

where $\gamma$ is the learning rate, $J$ is the loss function, and $\nabla_x J$ is the gradient of the loss function w.r.t. the input $x$.

We hereby introduce this gradient normalization to visual prompting. We define the matrix representation of EVP as $V_e = W \odot M$, where $W \in \mathbb{R}^{K \times K \times 3}$ are the prompt parameters, $M \in \mathbb{R}^{K \times K \times 3}$ is the mask matrix, and $\odot$ denotes the element-wise matrix product. The mask matrix $M$ encodes the spatial locations of the prompts, with the central part of size $k \times k$ being all zeros and the rest being all ones. In practice, we find that dividing the gradient of EVP by the $L_2$ norm of the gradient of $W$ leads to the best performance:

$$V_e^{t+1} = V_e^t - \gamma \frac{\nabla_{V_e^t} J}{||\nabla_W J||_2} \, . \tag{2}$$

We provide more details and ablation results on different normalization strategies in Section 5.3.

## 4 Experiments

**Datasets.** We evaluate visual prompting methods on 12 downstream classification datasets, including CIFAR100, CIFAR10 (Krizhevsky et al., 2009), Flowers102 (Nilsback & Zisserman, 2008), Food101 (Bossard et al., 2014), EuroSAT (Helber et al., 2019), SUN397 (Xiao et al., 2010), SVHN (Netzer et al., 2011), DTD (Cimpoi et al., 2014), OxfordPets (Parkhi et al., 2012), Resisc45 (Cheng et al., 2017), CLEVR (Johnson et al., 2017), and DMLab (Beattie et al., 2016). In addition, we test the robustness of visual prompting on 3 out-of-distribution datasets (Koh et al., 2021) (Camelyon17, FMoW, and iWildCAM), and 2 corruption datasets (Hendrycks & Dietterich, 2018) (CIFAR100-C and CIFAR10-C).

**Baselines.** We compare the performance of EVP with other commonly used prompting methods and fine-tuning protocols, including TP (text prompting), VP, VPT, LP (linear probing), and FT (fully fine-tuning). Specifically, we should note 1) TP is equivalent to zero-shot in CLIP; 2) LP uses a linear layer as the classification head; and 3) FT updates all parameters of the backbone and the classification head.

### 4.1 CLIP

Following the protocol of VP (Bahng et al., 2022), we conduct evaluations using the CLIP-Base/32 model on 12 classification datasets. The full results are shown in Table 13 and a detailed comparison to the two strong baselines, LP and VPT, is presented in Figure 3. Our proposed EVP approach consistently outperforms all previous prompting methods, with similar or fewer parameters. On average, EVP shows an improvement of 6.0% over VP, and 5.4% over VPT.

Table 1: Performance comparison across 12 datasets with CLIP. EVP substantially beats other visual prompting methods by a large margin. More notably, EVP outperforms the linear probing on 7 out of 12 datasets with a similar number of parameters. The results where EVP outperforms linear probing are highlight in **bold**.

| Adaptation | Tunable params (M) | CIFAR100 | CIFAR10 | Flowers | Food | EuroSAT | SUN | DMLab | SVHN | Pets | DTD | RESISC | CLEVR | Avg. |
|---|---|---|---|---|---|---|---|---|---|---|---|---|---|---|
| TP | 0 | 63.1 | 89.0 | 61.8 | 83.2 | 34.1 | 58.0 | 30.2 | 11.0 | 85.9 | 42.8 | 42.4 | 20.2 | 51.8 |
| VP | 0.070 | 75.3 | 94.2 | 62.0 | 83.2 | 95.6 | 68.4 | 41.9 | 88.4 | 86.5 | 57.1 | 84.1 | 81.4 | 76.5 |
| VPT | 0.064 | 76.6 | 95.0 | 76.2 | 84.7 | 94.6 | 69.3 | 48.4 | 86.1 | 92.1 | 61.6 | 84.3 | 58.6 | 77.3 |
| EVP(Ours) | 0.062 | **81.2** | **96.6** | 82.3 | 84.1 | **97.6** | 71.0 | **62.3** | 90.5 | **90.0** | 68.4 | 89.7 | **75.9** | 82.5 |
| LP | 0.037 | 80.0 | 95.0 | 94.1 | 88.3 | 94.8 | 76.2 | 49.3 | 65.4 | 89.2 | 73.5 | 92.3 | 66.1 | 80.3 |
| FT | 151.28 | 82.1 | 95.8 | 97.4 | 87.8 | 99.0 | 79.0 | 63.5 | 95.7 | 88.5 | 72.3 | 98.1 | 94.4 | 88.1 |

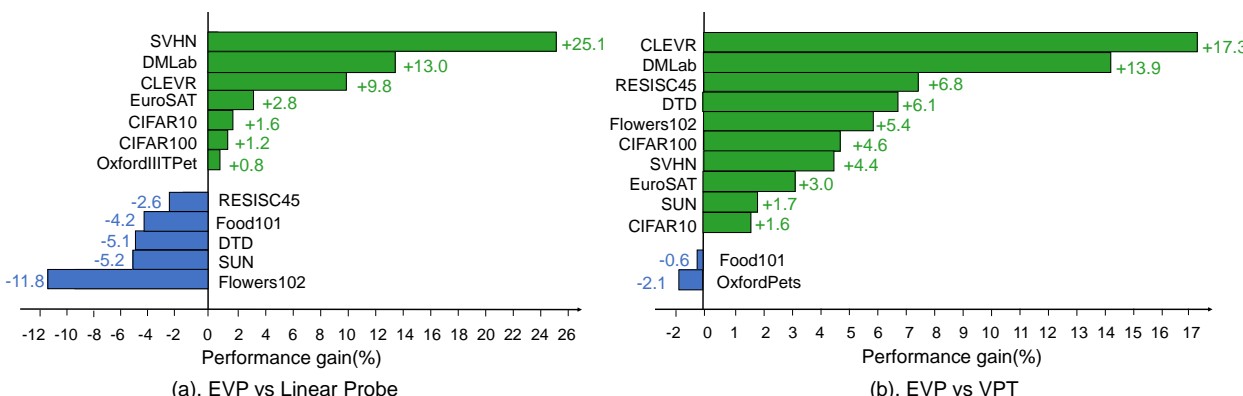

(a). EVP vs Linear Probe    (b). EVP vs VPT

Figure 3: **Performance gain** of EVP compared to linear probing and VPT on each downstream dataset. The bars indicate the gain or loss in accuracy compared to linear probing and VPT, respectively. (a) Compared with linear probing, EVP outperforms linear probing on 7 out of 12 datasets by **2.1%** on average. (b) Compared with VPT, EVP beats VPT on 10 out of 12 datasets by **5.6%** on average.

Table 2: **Performance of non-CLIP models**. EVP[*] indicates that we train EVP using classes after preprocessing stage. EVP slightly exceeds VP and EVP[*] outperforms EVP and VP by a large margin. The **bold** indicates the cases that the performance of EVP[*] is competitive with linear probing.

| Model | Adaptation | CIFAR100 | CIFAR10 | Flowers | Food | EuroSAT | SUN | SVHN | Pets | DTD | RESISC | CLEVR | Avg. |
|---|---|---|---|---|---|---|---|---|---|---|---|---|---|
| Instagram | VP | 16.7 | 62.1 | 4.8 | 6.5 | 86.1 | 2.2 | 53.8 | 18.6 | 29.1 | 40.6 | 30.9 | 31.9 |
| Instagram | EVP | 13.6 | 67.2 | 9.2 | 7.1 | 85.6 | 7.9 | 50.8 | 16.3 | 29.0 | 38.0 | 48.1 | 33.9 |
| Instagram | EVP[*] | **60.3** | **93.5** | 11.4 | 8.4 | 88.7 | 19.6 | **55.3** | 74.4 | 44.4 | 47.5 | **50.5** | 50.4 |
| Instagram | LP | 64.0 | 90.1 | 92.7 | 65.8 | 95.5 | 58.1 | 48.0 | 94.5 | 70.9 | 95.7 | 30.2 | 73.2 |
| Instagram | FT | 77.8 | 77.8 | 94.5 | 75.6 | 97.4 | 56.7 | 96.8 | 93.9 | 73.5 | 93.4 | 87.9 | 84.1 |
| RN50 | VP | 10.1 | 54.5 | 4.7 | 5.1 | 80.7 | 1.1 | 57.1 | 10.8 | 8.2 | 28.3 | 29.5 | 26.4 |
| RN50 | EVP | 9.2 | 55.9 | 6.6 | 3.9 | 85.5 | 5.1 | 48.6 | 10.5 | 18.7 | 35.4 | 35.5 | 28.6 |
| RN50 | EVP[*] | 24.9 | 77.0 | 11.9 | 7.0 | 81.0 | 14.7 | 47.8 | 72.0 | 41.2 | 39.2 | **37.2** | 41.3 |
| RN50 | LP | 67.7 | 87.7 | 92.7 | 62.5 | 95.8 | 57.5 | 60.3 | 91.1 | 66.7 | 92.2 | 32.6 | 73.3 |
| RN50 | FT | 79.9 | 94.1 | 96.9 | 73.2 | 96.5 | 55.9 | 96.9 | 92.3 | 66.7 | 93.4 | 89.3 | 84.3 |

To further evaluate the effectiveness, we next compare EVP with linear probing, which is a widely used fine-tuning protocol. The results, shown in Table 13 and Figure 3, demonstrate that EVP outperforms linear probing on 7 out of 12 datasets. On average, EVP achieved an accuracy of 82.5%, which is 2.2% higher than linear probing. In addition, our method is more flexible as the number of parameters can be easily controlled (via adjusting the padding size), whereas the number of parameters in linear probing must depend on the number of class categories in the downstream tasks.

Lastly, Our method, EVP, exhibits promising performance compared to fully fine-tuning while being significantly more parameter-efficient, with only 0.04% of the number of parameters. While there is still a performance gap between these two methods, with an average accuracy of 82.5% for EVP and 88.1% for fully fine-tuning, EVP outperforms or achieves similar results on certain datasets, including CIFAR100, CIFAR10, EuroSAT, DMLab, and Pets.

Table 3: Robustness comparison on **out-of-distribution** and **corruption** datasets. Left: out-of-distribution datasets. Right: corruption datasets. We can observe that EVP achieves much stronger robustness on both out-of-distribution setting and corruption setting.

| Model | Adaptation | iwildcam | camelyon17 | fmow | Avg. |
|-------|-----------|----------|------------|------|------|
| CLIP  | VP        | 57.3     | 91.4       | 37.8 | 62.2 |
| CLIP  | VPT       | 58.8     | 91.9       | 29.7 | 60.1 |
| CLIP  | EVP(Ours) | 64.9     | **95.1**   | 40.2 | **66.7** |
| CLIP  | LP        | **66.7** | 86.0       | 36.3 | 63.0 |
| CLIP  | FT        | 64.0     | 84.3       | **49.7** | 66.0 |

| Model | Adaptation | CIFAR100-C | CIFAR10-C | Avg. |
|-------|-----------|------------|-----------|------|
| CLIP  | VP        | 52.5       | 78.3      | 65.4 |
| CLIP  | VPT       | 54.0       | 70.2      | 62.3 |
| CLIP  | EVP(Ours) | 58.6       | **84.3**  | 71.5 |
| CLIP  | LP        | 56.9       | 78.8      | 67.9 |
| CLIP  | FT        | **61.1**   | 82.7      | 71.9 |

## 4.2 Non-CLIP Models

In this section, we evaluate the effectiveness of EVP on non-CLIP models. One challenge for adapting non-CLIP models to downstream tasks is that their original classification head is either less semantically meaningful or mapped to a set of predefined classes. A direct and naive solution used in VP is to arbitrarily map downstream classes to pre-trained classes and discard all unassigned classes. However, we posit that there could exist some similarity between pre-trained and downstream classes, even if an exact correspondence is not known. This motivates us to propose a pre-processing stage before implementing visual prompting to utilize this potential similarity.

Specifically, for each downstream class, we feed downstream images in that class into the pre-trained model and investigate the predictions in the pre-trained classes. We then simply choose the pre-trained class with the highest prediction frequency as the corresponding class for the downstream class. After pre-processing, we fix the correspondence and train our visual prompting method.

Overall, the results in Table 2 demonstrate the effectiveness of our proposed EVP method for adapting non-CLIP models to downstream tasks. While the performance of EVP is already improved over the baseline VP method (by 2%) when using arbitrary mapping, the use of our pre-processing stage substantially enhances the performance of EVP further, *i.e.*, from 33.9% to 50.4% with the Instagram pretraining model (Mahajan et al., 2018) and from 28.6% to 41.3% with an ImageNet-train ResNet-50. However, we note the performance of EVP$^*$ is not as strong on fine-grained datasets, such as Flowers102 and Food101, suggesting that it may be more challenging to find correspondence between pre-trained and downstream classes for these types of tasks.

## 4.3 Robustness

In this section, we investigate the robustness of EVP compared with other prompting methods and fine-tuning protocols. We evaluate the robustness of EVP on out-of-distribution (OOD) and corruption datasets.

**OOD robustness.** We test the robustness of EVP to distribution shift using the WILDS benchmark (Koh et al., 2021). The model is trained on datasets from a specific domain and then evaluated on datasets from a different domain, such as images from different regions, cameras, and hospitals. The results in Table 14 show that EVP outperforms other prompting methods by at least 4.5%. Additionally, we find that EVP outperforms both linear probing (+3.7%) and fully fine-tuning (+0.7%) in this setting, highlighting the potential of EVP in handling out-of-distribution samples.

**Robutness on corruption datasets.** In this study, we also evaluate the robustness of EVP to common image corruptions (Hendrycks & Dietterich, 2018). We test EVP on the CIFAR100-C and CIFAR10-C corruption datasets, which apply 19 common image corruptions to the CIFAR100 and CIFAR10 datasets, respectively. We train EVP on the CIFAR100 and CIFAR10 datasets and then evaluate its performance on the corresponding corruption datasets.

The average accuracy is reported in Table 14 (the accuracy under each type of corruption is reported in the supplementary material), where we can observe that EVP outperforms other prompting methods and linear probing by a large margin. It is also worth noting that EVP performs comparably to fully fine-tuning in

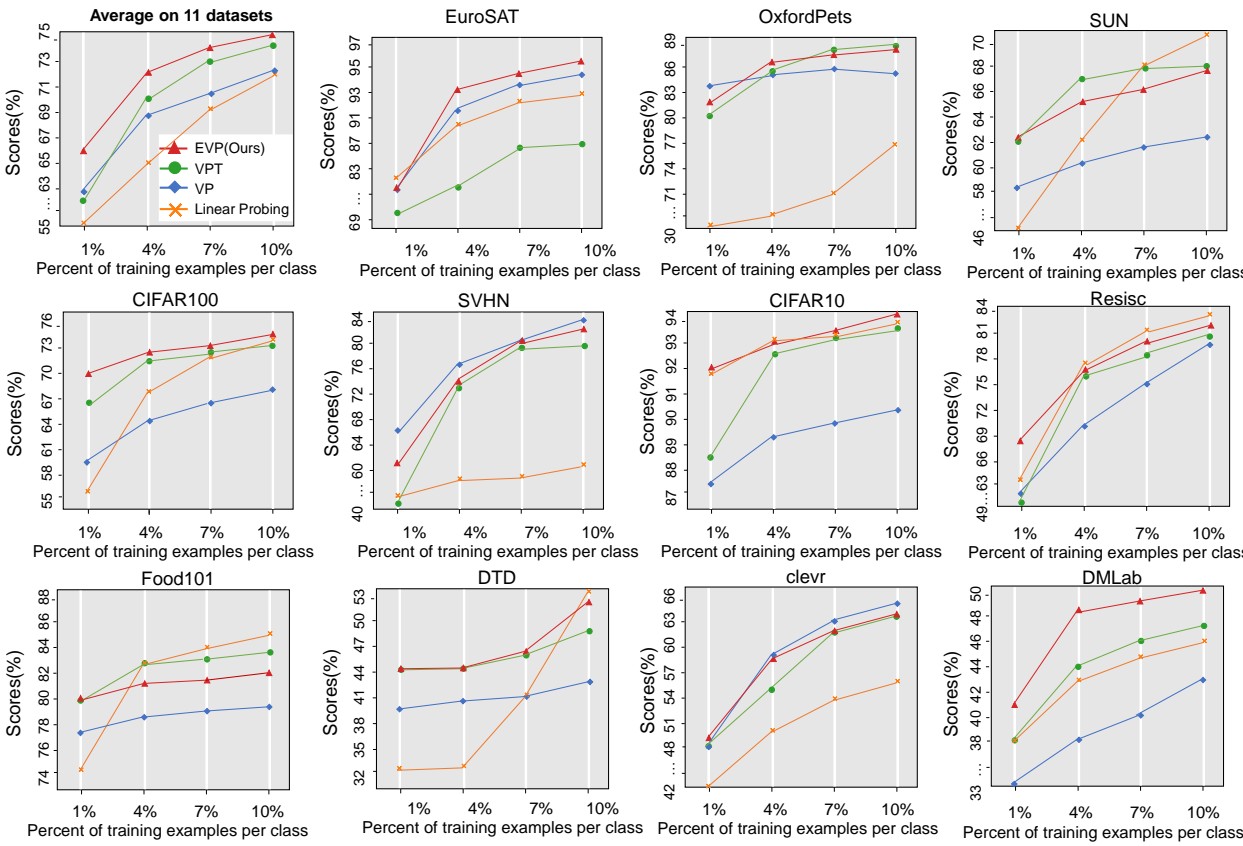

Figure 4: **Results with different data scales** on 11 visual recognition datasets. Each figure shows the results trained on 1%, 4%, 7%, and 10% data, respectively. All visual prompting methods show clear dominance compared with linear probing. EVP (red line) outperforms other methods by a large margin on average.

handling common image corruptions, *i.e.*, 71.5% *vs.* 71.9%. This may be due to the fact that the corruptions on the image can damage the performance of other baselines, while our strategy of treating the prompt as a standalone and independent component can alleviate this issue, *i.e.*, the prompts in EVP are not directly combined with (but are padded around) the corrupted images during inference.

## 4.4 Different Data Scales

In this section, we evaluate the performance of EVP with different data scales. We train EVP using only 1%, 4%, 7%, and 10% of the data for each class in the training datasets. This is of particular interest because few-shot learning is an important aspect of prompting in natural language processing. We hereby aim to validate if EVP can achieve strong performance with limited data.

The average accuracy, as well as the detailed accuracy on each dataset, is reported in Figure 4. We can observe that 1) visual prompting methods (VP, VPT, EVP) consistently outperform linear probing, demonstrating the effectiveness of visual prompting in learning with limited labeled data; and 2) among all visual prompting methods, EVP consistently achieves the best overall performance, demonstrating its strong generalization ability at different data scale.

## 5 Discussion

### 5.1 Original Image Information

We first investigate the importance of preserving the original information of the input image. To explore this, we manipulate the size of the input image while keeping the number of learnable parameters constant.

Table 4: **Ablation on original image information** across 12 datasets on CLIP. There is a clear trend that as the image size decreases, *i.e.*, as the occlusion decreases, the performance gradually increases.

| Image size | CIFAR100 | CIFAR10 | Flowers | Food | EuroSAT | SUN | DMLab | SVHN | Pets | DTD | RESISC | CLEVR | Avg. |
|---|---|---|---|---|---|---|---|---|---|---|---|---|---|
| 224 | 77.1 | 94.7 | 80.6 | 81.8 | 96.7 | 71.4 | 60.1 | 88.9 | 89.1 | 64.2 | 88.1 | 73.4 | 80.5 |
| 204 | 77.5 | 95.3 | 80.8 | 82.0 | 97.0 | 71.0 | 60.7 | 89.7 | 89.3 | 64.0 | 88.5 | 73.6 | 80.8 |
| 184 | 79.0 | 95.9 | 81.7 | 82.3 | 97.4 | 70.3 | 61.9 | **90.6** | **89.9** | 65.2 | 89.5 | 74.3 | 81.5 |
| 164 (Default) | **81.2** | **96.6** | **82.3** | **82.3** | **97.6** | 71.0 | **62.3** | 90.5 | 88.7 | **68.4** | **89.7** | **75.9** | 82.2 |

Table 5: **Ablation on the positional embedding at the pixel level**. EVP-small shrinks the original image and pads it with learnable pixels to the original size, while EVP-big pads pixel patches around the original image. We note EVP-small w/ PE can beat EVP-big w/o PE despite having fewer parameters and a smaller input resolution, suggesting positional embeddings are crucial in visual prompting at the pixel level.

| Image size | CIFAR100 | CIFAR10 | Flowers | Food | EuroSAT | SUN | SVHN | Pets | DTD | RESISC | CLEVR | Avg. |
|---|---|---|---|---|---|---|---|---|---|---|---|---|
| EVP-small w/ PE | 81.2 | 96.6 | 82.3 | 84.1 | 97.6 | 71.0 | 90.5 | 90.0 | 68.4 | 89.7 | 75.9 | 84.3 |
| EVP-small w/o PE | 70.9 | 92.3 | 70.2 | 78.6 | 83.0 | 62.2 | 55.9 | 88.1 | 62.5 | 73.8 | 72.1 | 73.6 |
| EVP-big w/ PE | 81.4 | 96.9 | 92.3 | 84.7 | 97.4 | 71.8 | 90.7 | 89.2 | 68.9 | 91.6 | 77.1 | 85.6 |
| EVP-big w/o PE | 73.4 | 93.7 | 71.0 | 80.7 | 85.3 | 63.4 | 58.1 | 88.9 | 64.6 | 76.2 | 74.5 | 75.4 |

Table 6: **Ablation on the positional embedding at the token level**. VPT only adds positional embeddings to the image patch embeddings, while $VP_1T$, $VP_{25}T$, $VP_{50}T$ denote methods in which the 1st, 25th, and 50th positional embeddings are added to the learnable tokens, respectively. We can observe that simply adding positional embeddings to the learnable tokens can significantly improve performance.

| Image size | CIFAR100 | CIFAR10 | Flowers | Food | EuroSAT | SUN | SVHN | Pets | DTD | RESISC | CLEVR | Avg. |
|---|---|---|---|---|---|---|---|---|---|---|---|---|
| VPT | 76.6 | 95.0 | 76.2 | 84.7 | 94.6 | 69.3 | 86.1 | 92.1 | 61.6 | 84.3 | 58.6 | 79.9 |
| $VP_1T$ | 77.3 | 96.0 | 77.5 | 84.9 | 96.2 | 69.7 | 87.3 | 92.2 | 67.7 | 87.1 | 59.9 | 81.4 |
| $VP_{25}T$ | 76.8 | 95.5 | 76.3 | 84.0 | 95.9 | 69.0 | 85.4 | 92.0 | 66.6 | 86.1 | 58.8 | 80.6 |
| $VP_{50}T$ | 77.0 | 96.0 | 75.4 | 83.8 | 95.8 | 69.3 | 86.1 | 92.3 | 66.4 | 84.0 | 59.0 | 80.4 |

By scaling the size of the input image beyond 164, we expect to see an increased overlap between the input image and the prompt. The results are reported in Table 4. A notable observation is that for 9 out of the 12 datasets examined, the performance is inversely proportional to the extend of overlap. This underscores the importance of preserving original image information.

## 5.2 Prompting Positional Embedding

We next probe positional embeddings. Note while EVP encodes the positions of both learnable visual prompts and image patch embeddings, VPT restricts positional embeddings (**PE**) solely for the image patch embeddings. To illuminate the implications of positional embeddings in visual prompting, we present evaluations across different configurations, both at the pixel and token levels.

Focusing on the pixel level, we denote our main method as **EVP-small w/ PE**, and introduce two variations: 1) **EVP-big w/ PE**: Here, the image at the original size is padded with learnable pixels, and the positional embeddings are added to both the image patch embeddings and the learnable pixels using interpolation. 2) **EVP-big w/o PE**: we hereby also use the image at the original size, but only add positional embeddings to the image patch embeddings. Table 5 shows the results. We can observe **EVP-big w/ PE** achieves the best performance, while **EVP-big w/o PE** performs the worst. For example, we note even **EVP-small w/ PE** is able to outperform **EVP-big w/o PE** by an average of 6.1% on the five datasets (86.7% *vs.* 78.6%), despite having fewer parameters and lower image resolution. These results demonstrate the vital role of adding positional embeddings in visual prompting.

At the token level, we find that simply adding positional embeddings to the learnable tokens can improve performance. To investigate this further, we create different prompting choices at the token level by adding different positional embeddings to the learnable tokens. Specifically, we denote prompting choices as $VP_nT$, where the $n$-th positional embeddings are added to the learnable tokens, as illustrated in Figure 5. Since the

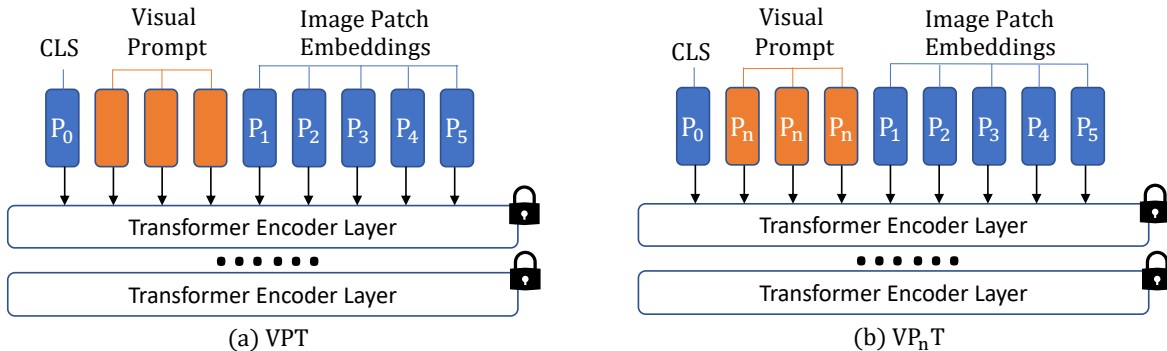

Figure 5: **Ablation on positional embedding at the token level.** (a). Visual Prompting Tuning(VPT): Inject learnable tokens between CLS and image patch embedding without positional embedding (b): $VP_nT$: Inject learnable tokens between CLS and image patch embedding with same positional embedding $P_n$ (i.e, $n$-th positional embedding (n = 1, 2, ..., 5)).

Table 7: Ablation on **augmentation** and **normalization**. Left panel: We find that simple techniques like RandomFlip and RandomCrop achieve strong results, while stronger augmentations like Cutmix or RandAug decrease the performance. Right panel: We note 1) applying L2 norm on gradient enhance performance; and 2) using the whole image's gradient leads to further improvement compared to using only the gradient of the visual prompting pixels.

| Augmentation | | | Performance |
|---|---|---|---|
| Flip&crop | RandAug | CutMix | |
| ✗ | ✗ | ✗ | 80.5 |
| ✔ | ✗ | ✗ | **81.2** |
| ✔ | ✔ | ✗ | 79.4 |
| ✔ | ✗ | ✔ | 79.7 |

| Gradient Normalization | | | | Performance |
|---|---|---|---|---|
| $L_1$ | $L_\infty$ | $L_2$-partial | $L_2$-whole | |
| ✗ | ✗ | ✗ | ✗ | 77.5 |
| ✔ | ✗ | ✗ | ✗ | 77.2 |
| ✗ | ✔ | ✗ | ✗ | 71.9 |
| ✗ | ✗ | ✔ | ✗ | 79.4 |
| ✗ | ✗ | ✗ | ✔ | **81.2** |

total token count is 50, in our experiment, we choose $n = \{1, 25, 50\}$ to study the effects of inserting tokens at the head, the middle, and the tail of the original image.

The results in Table 6 demonstrate that the incorporation of positional embeddings in the learnable tokens can consistently and significantly improve the performance of visual prompting. For example, the average accuracy is increased by $+2.4\%$ with $VP_1T$, $+1.8\%$ with $VP_{25}T$, and $+1.4\%$ with $VP_{50}T$. Collectively, these findings reiterate the importance of positional embeddings in this context.

## 5.3 Training strategy

In this section, we turn our attention to the impact of various "old tricks" in transferable adversarial learning on the performance of visual prompting. We first ablate different augmentation methods such as RandomHorizontalFlip, RandAug, and Cutmix on the CIFAR100 dataset. As shown in Table 7, interestingly, we find that using simple augmentation techniques like RandomHorizontalFlip and RandomCrop can already achieve satisfactory results, while more advanced methods such as Cutmix or RandAug may decrease performance, likely due to over-regularization. For example, on CIFAR100, RandomHorizontalFlip and RandomCrop is able to improve accuracy by 0.7%, but RandAug or CutMix hurts the accuracy by 1.1% and 0.8%, respectively.

Next, we ablate different gradient normalization strategies, including the $L_1$ norm, $L_2$ norm, and $L_\infty$ norm, on the CIFAR100 dataset. The results, shown in Table 7, indicate that the $L_2$ norm consistently performs the best among all three strategies. Furthermore, we observe that employing the entirety of the image's gradient to compute the norm ($L_2$-whole) performs consistently better than merely using the gradient of the visual prompting pixels ($L_2$-partial), *i.e.*, 81.2% *vs.* 79.4%.

Table 8: **Performance of ViT models**. By incorporating the output transform into the framework, we can observe a substantial improvement.

| Model | Adaptation | CIFAR100 | CIFAR10 | Flowers | Food | EuroSAT | SUN | SVHN | Pets | DTD | RESISC | CLEVR | Avg. |
|-------|-----------|----------|---------|---------|------|---------|-----|------|------|-----|--------|-------|------|
| ViT | VP | 44.6 | 94.9 | 15.3 | 36.5 | 95.3 | 2.2 | 16.2 | 8.8 | 16.8 | 64.8 | 35.7 | 39.2 |
| ViT | VP+ILM | 73.8 | 97.0 | 36.9 | 42.6 | 96.0 | 17.6 | 89.7 | 79.5 | 36.2 | 69.6 | 64.0 | 64.4 |
| ViT | EVP | 73.8 | 97.7 | 78.6 | 62.7 | 97.6 | 5.7 | 19.6 | 8.7 | 57.4 | 89.5 | 50.2 | 58.3 |
| ViT | EVP+ILM | 86.3 | 97.6 | 77.5 | 70.3 | 97.6 | 26.2 | 95.2 | 86.6 | 60.1 | 89.8 | 71.2 | 78.0 |

## 5.4 VPT-DEEP

VPT-DEEP is an advanced version of VPT, which additionally introduces learnable tokens at every Transformer block for enhancing performance. Following the setup in Section 4, we hereby draw a comparative performance analysis between VPT-DEEP and EVP, focusing on the following three settings: the CLIP-based model, out-of-distribution scenarios, and data corruption scenarios. Regarding CLIP-based model, our results show that, while VPT-DEEP outperforms the vanilla VPT by 4.6% (from 77.3% to 81.9%) with significantly more parameters at 0.092 million, its performance remains slightly lower than that of our EVP, which achieves an average accuracy of 82.5%. In addition, EVP consistently achieves superior performance than VPT-DEEP in more challenging settings, specifically, out-of-distribution (66.7% *vs.* 65.2%) and corruption (71.5% *vs.* 69.5%). These findings confirm the efficacy of EVP as a superior prompting strategy. For more detailed accuracy scores across individual datasets, please refer to the supplementary material.

## 5.5 Compatibility with label mapping strategy

Though this work only focuses on optimization on input pixels, it is also compatible with other techniques. For examples, label mapping strategy is an effective method in visual prompting, which automatically maps the source labels to target domain labels, in order to mitigate the uncalibrated final layer (i.e., classification head) of non-CLIP models. In order to further enhance the performance of visual prompting, this section delves into the in-depth exploration of the compatibility between EVP and ILM. Specifically, we conducted experiments under ViT-Base 16 as the backbone, and results are shown in Tab. 8. We can observe that the integration of ILM results in a substantial performance improvement of both VP and EVP. In the case of VP and EVP, ILM achieved performance improvement of 25.2% (39.2% to 64.4%) and 19.7% (58.3 % to 78.0 %) respectively. Moreover, irrespective of the use of ILM, EVP consistently outperforms VP.

## 6 Conclusion

We propose EVP, a simple and effective method for adapting pre-trained models to various downstream tasks using visual prompts at the pixel level. EVP preserves the original image information and incorporate adversarial learning techniques to improve performance. Our experiments demonstrate that EVP outperforms other visual prompting methods and outperforms linear probing in a variety of settings. Moreover, EVP shows strong performance in handling data of varying scales and robustness against out-of-distribution samples.

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

## A    Implementation details

Our implementation is based on Pytorch (Paszke et al., 2019). We use CLIP-B/32, Instagram (Mahajan et al., 2018), and ResNet50 (He et al., 2016) as our pre-trained model, and the batch size is 256, 32, 128, respectively. All visual prompting in our experiments are trained for 1000 epochs. For EVP, we use SGD with a cosine learning rate schedule; the initial learning rate is 70. The prompting size is 30 pixels by default. To fairly compare with VP, we follow its text prompting setup (Bahng et al., 2022) in CLIP model. Specifically, we use "This is a photo of a [LABEL]" by default for the text prompting. For CLEVR datasets, we use "This is a photo of [LABEL] objects", for DMLab datasets, we use "The distance is [LABEL1], and the reward is [LABEL2]", and for Camelyon17, the text prompting template is "a tissue region [LABEL] tumor".

## B    Prompting size

The prompting size is defined as $p = \frac{K-k}{2}$, where k is the image size after shrinking, and K is the input size of pre-trained model. Therefore, the number of parameters is $12p(K - p)$, which only depends on p since K is fixed for a given model. In our experiment, the optimal prompting size varies across datasets, as shown in Figure 6. Since we shrink the original image and pad learnable pixels around it, there shows a trade-off between the image resolution and the number of parameters. Interestingly, we note that, for datasets with a low resolution (*e.g.*, CIFAR100), the prompting of $p = 30$ achieves the best performance. While for datasets with a high resolution, we note setting p to a small value empirically works the best. For example, we find that $p = 5$ is the best in Food101 dataset which has a resolution of $512 \times 512$. The best hyper-parameter is shown in Table 12 and we note setting prompting size to 30 generally achieve the best overall accuracy on these four datasets.

Table 9: **The optimal prompting size in our experiments** across 12 datasets on CLIP.

| Image size | CIFAR100 | CIFAR10 | Flowers | Food | EuroSAT | SUN | DMLab | SVHN | Pets | DTD | RESISC | CLEVR |
|---|---|---|---|---|---|---|---|---|---|---|---|---|
| Prompt size (p) | 30 | 30 | 30 | 5 | 30 | 30 | 30 | 30 | 20 | 30 | 30 | 30 |

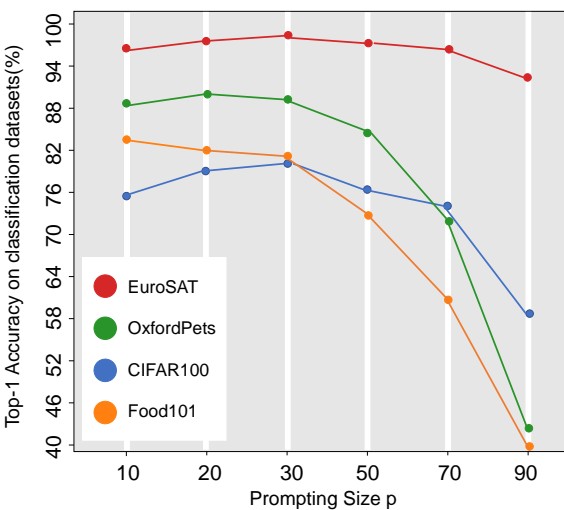

Figure 6: **Ablation on prompting size.** The pre-trained model is CLIP-B/32. We vary the prompting size, which determines the number of parameters, and show the performance on four datasets.

Table 10: Specific performance on CIFAR-10C

| Methods | brightness | contrast | defocus_blur | elastic | fog | frost | gaussian_blur | gaussian_noise | glass_blur | impulse_noise |
|---|---|---|---|---|---|---|---|---|---|---|
| VP | 91.9 | 81.6 | 87.8 | 82.0 | 86.5 | 84.8 | 85.4 | 61.4 | 60.7 | 61.9 |
| VPT | 87.6 | 78.0 | 82.0 | 73.9 | 78.8 | 77.1 | 79.9 | 46.8 | 46.0 | 55.8 |
| EVP(Ours) | 95.4 | 90.3 | 93.1 | 87.7 | 91.5 | 89.8 | 91.6 | 66.3 | 69.0 | 67.8 |
| LP | 92.9 | 85.0 | 88.6 | 82.1 | 87.0 | 85.6 | 86.2 | 56.2 | 55.1 | 62.1 |
| FT | 94.9 | 91.6 | 92.1 | 84.2 | 91.2 | 88.5 | 91.1 | 57.1 | 59.5 | 68.7 |

| Methods | jpeg_compression | motion_blur | pixelate | saturate | shot_noise | snow | spatter | speckle_noise | zoom_blur | Avg. |
|---|---|---|---|---|---|---|---|---|---|---|
| VP | 74.3 | 79.7 | 66.6 | 88.9 | 67.5 | 84.8 | 87.3 | 69.1 | 84.6 | 78.2 |
| VPT | 63.9 | 72.5 | 58.1 | 84.1 | 56.1 | 77.9 | 80.6 | 59.0 | 75.9 | 70.2 |
| EVP(Ours) | 80.4 | 86.3 | 75.3 | 93.2 | 73.6 | 89.6 | 91.0 | 75.2 | 90.3 | 84.3 |
| LP | 75.1 | 80.8 | 77.9 | 90.5 | 65.0 | 86.2 | 88.6 | 67.8 | 85.2 | 78.8 |
| FT | 74.3 | 86.8 | 77.3 | 93.0 | 66.4 | 89.0 | 91.0 | 69.4 | 89.0 | 82.7 |

Table 11: Specific performance on CIFAR-100C

| Adaptation | brightness | contrast | defocus_blur | elastic | fog | frost | gaussian_blur | gaussian_noise | glass_blur | impulse_noise |
|---|---|---|---|---|---|---|---|---|---|---|
| VP | 71.3 | 55.5 | 65.7 | 57.9 | 61.5 | 58.0 | 62.2 | 30.6 | 27.6 | 37.4 |
| VPT | 72.8 | 62.8 | 67.9 | 57.9 | 64.6 | 58.7 | 65.4 | 30.3 | 25.8 | 41.0 |
| EVP(Ours) | 77.7 | 66.5 | 73.2 | 63.5 | 69.1 | 64.2 | 70.2 | 36.3 | 31.8 | 38.5 |
| LP | 75.5 | 63.1 | 69.8 | 60.0 | 66.2 | 63.4 | 66.4 | 31.1 | 30.8 | 39.4 |
| FT | 80.7 | 73.1 | 75.7 | 62.9 | 72.9 | 66.6 | 73.6 | 34.8 | 31.9 | 46.1 |

| Adaptation | jpeg_compression | motion_blur | pixelate | saturate | shot_noise | snow | spatter | speckle_noise | zoom_blur | Avg. |
|---|---|---|---|---|---|---|---|---|---|---|
| VP | 47.4 | 55.2 | 46.1 | 61.7 | 37.7 | 59.3 | 63.5 | 38.6 | 61.1 | 52.5 |
| VPT | 45.2 | 57.8 | 45.0 | 64.3 | 38.0 | 60.5 | 64.9 | 39.4 | 63.4 | 54.0 |
| EVP(Ours) | 52.8 | 63.5 | 51.4 | 69.1 | 43.8 | 65.4 | 68.6 | 44.6 | 68.0 | 58.6 |
| LP | 49.5 | 61.4 | 57.2 | 66.8 | 39.7 | 64.7 | 67.8 | 42.4 | 65.8 | 56.9 |
| FT | 48.3 | 66.5 | 52.6 | 73.5 | 42.9 | 69.1 | 72.9 | 44.3 | 70.9 | 61.1 |

# C  Comparative Analysis of Model Size, Training Time, and Throughput

In order to provide a more comprehensive and in-depth comparison, we conducted experiments on CIFAR100 and listed the training time, throughput, and the number of parameters of vp, vpt, evp, and linear probe, respectively. showed the results.

Table 12: Model Size, Training Time, and Throughput of different methods

| Adaptation | Number of parameters (Millions) | Training time (mins) | Throughout (images/second) |
|---|---|---|---|
| VP | 0.070 | 358 | 780 |
| VPT | 0.064 | 1050 | 651 |
| EVP | 0.062 | 409 | 961 |
| Linear probing | 0.051 | 356 | 1130 |

Table 13: Performance comparison across 12 datasets with CLIP. We note EVP outperforms VPT-DEEP with fewer tunable parameters. The results where EVP outperforms VPT-DEEP are highlight in **bold**.

| Adaptation | Tunable params (M) | CIFAR100 | CIFAR10 | Flowers | Food | EuroSAT | SUN | DMLab | SVHN | Pets | DTD | RESISC | CLEVR | Avg. |
|---|---|---|---|---|---|---|---|---|---|---|---|---|---|---|
| VPT-DEEP | 0.092 | 78.3 | 96.1 | 84.4 | 85.6 | 97.4 | 70.2 | 57.7 | 90.1 | 92.5 | 70.1 | 90.6 | 69.7 | 81.9 |
| EVP(Ours) | 0.062 | **81.2** | **96.6** | 82.3 | 84.1 | **97.6** | **71.0** | **62.3** | **90.5** | 90.0 | 68.4 | 89.7 | **75.9** | 82.5 |

Table 14: Robustness comparison on **out-of-distribution** and **corruption** datasets. Left: out-of-distribution datasets. Right: corruption datasets. We can observe that EVP achieves much stronger robustness on both out-of-distribution setting and corruption setting.

| Model | Adaptation | iwildcam | camelyon17 | fmow | Avg. |
|---|---|---|---|---|---|
| CLIP | VPT-DEEP | 62.7 | 93.6 | 39.3 | 65.2 |
| CLIP | EVP(Ours) | 64.9 | **95.1** | 40.2 | **66.7** |

| Model | Adaptation | CIFAR100-C | CIFAR10-C | Avg. |
|---|---|---|---|---|
| CLIP | VPT-DEEP | 56.3 | 82.6 | 69.5 |
| CLIP | EVP(Ours) | 58.6 | **84.3** | 71.5 |

## D  Performance under different corruption cases

In Section 4.3, we see that our EVP outperforms other methods on corruption setting. Here, we list the generalization performance of all methods under various types of corruptions, as shown in Table 10 and Table 11. Specifically, compared to VP, we note 1) on CIFAR-10-C, EVP yields the largest improvement on constrast (+8.8%) and the smallest improvement on brightness(+3.5%); 2) on CIFAR-100-C, EVP yields the largest improvement on contrast (+11.0%) and the smallest improvement on impulse noise (+1.1%).

## E  Performance comparison with VPT-DEEP

VPT-DEEP is an advanced version of VPT, which additionally introduces learnable tokens at every Transformer layer's input space for enhancing performance. We hereby briefly compare its performance to that of EVP. Specifically, we compare the performance of EVP and VPT in three settings: CLIP-model, OOD, and corruption.

### E.1  Performance on CLIP-model

In this section, based on CLIP model, we conduct a comparative analysis of the performance of EVP and VPT-DEEP on 12 classification dataset. The results are shown in Table 13. We can see that EVP demonstrates an average performance improvement of 0.6% over VPT-DEEP(82.5% *v.s.* 81.9%), with only 0.062 Million tunable parameters, which is 0.03 million fewer than VPT-DEEP.

### E.2  Performance on Robustness

Following the setting in main text, we test the robustness of EVP and VPT-DEEP to distribution shift and common image corruption. Table 14 presents the specific comparative results. In both the OOD (+1.5%) and corruption settings (+2.0%), EVP achieves superior performance consistently compared to VPT-DEEP, which demonstrates the robustness of EVP.

## F   More ablation study results

In this section, we demonstrate more ablation study results of augmentation strategy and normalization strategy. As shown in Tab. 15, data augmentation and normalization play a great role in visual prompting.

Table 15: Ablation on augmentation and normalization. Simple techniques like RandomFlip and RandomCrop achieve strong results, and applying normalization on gradient enhance performance significantly.

| Adaptation | CIFAR100 | CIFAR10 | Flowers | Food | EuroSAT | SUN | SVHN | Pets | DTD | RESISC | CLEVR | Avg. |
|---|---|---|---|---|---|---|---|---|---|---|---|---|
| EVP w/o augmentation | 80.5 | 96.2 | 79.8 | 83.5 | 96.5 | 68.0 | 90.1 | 79.4 | 67.8 | 89.3 | 74.6 | 82.3 |
| EVP w/o normalization | 77.5 | 95.7 | 78.5 | 80.9 | 92.9 | 54.3 | 72.3 | 78.7 | 48.2 | 82.6 | 72.1 | 75.8 |
| EVP | 81.2 | 96.6 | 82.3 | 84.1 | 97.6 | 71.0 | 90.5 | 90.0 | 68.4 | 89.7 | 75.9 | 84.3 |

