# OpenReview forum: "Unleashing the Power of Visual Prompting At the Pixel Level"
_TMLR — Accepted by TMLR_

### Review · Reviewer_ExA8 · 2024-03-23

**Summary Of Contributions:**

This paper proposed EVP (Enhanced Visual Prompting): a visual prompting method that integrates non-overlapping images and learnable prompts. Strategies such as data augmentation and gradient normalization are employed to enhance performance. EVP's reported accuracy, robustness, and generalization ability on limited data surpass the baselines (VPT, VP, and LP). Additionally, the author identifies the significant role of position embedding in training visual prompts, making a comparison with VPT.

**Audience:**

Yes

**Claims And Evidence:**

Yes

**Requested Changes:**

Please refer to the weaknesses. In addition, please also address the following concerns:

1. The improvement in performance resulting from different techniques (augmentation and normalization in Sec. 5.3) has only been validated on CIFAR100. It is unclear whether these techniques would still effectively enhance accuracy on other datasets.

2. The discussion on different designs of positional embedding could be enhanced. In addition to interpolation and fixed n-th PE, exploring more optimal approaches could provide further insights.

**Strengths And Weaknesses:**

Strengths:
1. There are comprehensive experiments and comparisons on different configurations (image size, data size, gradient normalization).

2. The discussion on position embedding (PE) is enlightening, and the results show a substantial accuracy gap compared to the absence of PE.


Weaknesses:
1. The proposed method is straightforward. More explanation on why gradient normalization can benefit visual prompting should be provided.

2. In Table 5, only five datasets were selected in the experiments, but the reasons for not selecting the others were not provided.

3. It is not clear how the output transformation is being implemented. Recent works on Visual Prompting have also shown that prompt/image size [1, 2]  and output mapping [1, 3] could affect VP performance. A comparison with them will strengthen the proposed method.

4. As mentioned in Sec. 5.2, the results showed that simple augmentation techniques like RandomHorizontalFlip and RandomCrop can already achieve the best results. It is not clear why the proposed method considers other augmentation in the EVP framework.

5. The performance in non-CLIP models is much worse than LP (Table 2). The author could explore not only CNN-based models (i.e., ResNet-50 and Instagram) but also ViT.

6. The difference in trainable parameter size between EVP-small and EVP-big is not clearly explained. The statement "EVP-big pads pixel patches around the original image" in the caption of Table 5 is confusing; it is not clear whether the term "original image" refers to the dataset's resolution or the input size required by the model. Also, please also add the results of “EVP-small w/o PE.”

[1] AutoVP: An Automated Visual Prompting Framework and Benchmark (ICLR 2024)

[2] Exploring Visual Prompts for Adapting Large-Scale Models

[3] Understanding and improving visual prompting: A label-mapping perspective (CVPR 2023)

---

> ### Author Response · Authors · 2024-04-13
> **Re: Review of Paper2367 by Reviewer ExA8**
>
> Thank you for your valuable feedback on our paper. We appreciate the opportunity to clarify and discuss the issues in your comments.
>
> >**Question1:** The proposed method is straightforward. More explanation on why gradient normalization can benefit visual prompting should be provided.
>
> **Answer1:** The motivation for using gradient normalization is our observation of the similar formulations between visual prompts and adversarial attacks. Consequently, we endeavor to integrate gradient normalization within the framework of visual prompts.
> In the ablation study, we find it improves performance.  We postulate that the improvement may be attributed to the regulatory function enacted by gradient normalization.
>
>
> >**Question2:** In Table 5, only five datasets were selected in the experiments, but the reasons for not selecting the others were not provided.
>
> **Answer2:** Sorry for the confusion. Selecting those five datasets is purely driven by the limitation of space, and the selection process is random. Moreover, we present the full results below, where we can see that our analysis/conclusions still hold. We will add this full result in the next version.
>
>
> | | cifar100 | cifar10 | flower102 | food101 | eurosat | sun397  | svhn | pet  | dtd  | resisc | clevr | Avg.  |
> | ----- | -------- | ------- | ------ | ---- | ------- | ---- | ---- | ---- | ---- | ------ | ----- | ---- |
> | VPT   | 76.6     | 95.0    | 76.2   | 84.7 | 94.6    | 69.3 | 86.1 | 92.1 | 61.6 | 84.3   | 58.6  | 79.9 |
> | VP1T  | 77.3     | 96.0    | 77.5   | 84.9 | 96.2    | 69.7 | 87.3 | 92.2 | 67.7 | 87.1   | 59.9  | 81.4 |
> | VP25T | 76.8     | 95.5    | 76.3   | 84.0 | 95.9    | 69.0 | 85.4 | 92.0 | 66.6 | 86.1   | 58.8  | 80.6 |
> | VP50T | 77.0     | 96.0    | 75.4   | 83.8 | 95.8    | 69.3 | 86.1 | 92.3 | 66.4 | 84.0   | 59.0  | 80.4 |
>
>
>
>
> |                  | cifar100 | cifar10 | flower102 | food101 | eurosat | sun397  | svhn | pet  | dtd  | resisc | clevr | Avg.|
> | ---------------- | -------- | ------- | ------ | ---- | ------- | ---- | ---- | ---- | ---- | ------ | ----- | ---- |
> | EVP-small wo/ PE | 70.9     | 92.3    | 70.2   | 78.6 | 83.0    | 62.2 | 55.9 | 88.1 | 62.5 | 73.8   | 72.1  | 73.6 |
> | EVP-small w/ PE| 81.2     | 96.6    | 82.3   | 84.1 | 97.6    | 71.0 | 90.5 | 90.0 | 68.4 | 89.7   | 75.9  | 84.3 |
> | EVP-big w/ PE| 81.4     | 96.9    | 92.3   | 84.7 | 97.4    | 71.8 | 90.7 | 89.2 | 68.9 | 91.6   | 77.1  | 85.6 |
> | EVP-big wo/ PE   | 73.4     | 93.7    | 71.0   | 80.7 | 85.3    | 63.4 | 58.1 | 88.9 | 64.6 | 76.2   | 74.5  | 75.4 |
>
>
> >**Question3:** It is not clear how the output transformation is being implemented. Recent works on Visual Prompting have also shown that prompt/image size [1, 2] and output mapping [1, 3] could affect VP performance. A comparison with them will strengthen the proposed method.
>
> **Answer3:** Thanks for your suggestion. Our work diverges in its focal point from that of the ILM[1] and AutoVP[2], i.e., they focus on the output transform while ours focuses on the input pattern. In addition, we verify the compatibility of our method with the output transform proposed by ILM. By incorporating the output transform into the EVP framework, we can observe an improvement in the CLIP setting. More importantly, with the non-CLIP model (i.e., an ImageNet-21k pretrained ViT-B/16), this performance improvement is much more substantial, with specific results shown in the table below. We will add these interesting results in the next version.
>
>
>
> |              | cifar100 | cifar10 | flower102 | food101 | eurosat | sun397  | svhn | pet  | dtd  | resisc | clevr | Avg.  |
> | ------------ | -------- | ------- | ------ | ---- | ------- | ---- | -----  | ---- | ---- | ------ | ----- | ---- |
> | EVP_clip     | 81.2     | 96.6    | 82.3   | 84.1 | 97.6    | 71.0 |  90.5 | 90.0 | 68.4 | 89.7   | 75.9  | 82.5 |
> | EVP_clip+ILM | 80.8     | 96.4    | 85.9   | 83.8 | 97.4    | 72.6 | 91.2 | 89.0 | 66.2 | 88.7   | 77.0  | 82.7 |
>
>
>
> |            | cifar100 | cifar10 | flower102 | food101 | eurosat | sun397  | svhn | pet  | dtd  | resisc | clevr | Avg.  |
> | ---------- | -------- | ------- | ------ | ---- | ------- | ---- | ---- | ---- | ---- | ------ | ----- | ---- |
> | VP  | 44.6     | 94.9    | 15.3   | 36.5 | 95.3    | 2.2  | 16.2 | 8.8  | 16.8 | 64.8   | 35.7  | 39.2 |
> | EVP | 73.8     | 97.7    | 78.6   | 62.7 | 97.6    | 5.7  | 19.6 | 8.7  | 57.4 | 89.5   | 50.2  | 58.3 |
> | ILM+VP     | 79.3     | 97.0    | 36.9   | 42.6 | 96.0    | 17.6 | 89.7 | 79.5 | 36.2 | 69.6   | 64.0  | 64.4 |
> | ILM+EVP    | 86.3     | 97.6    | 77.5   | 70.3 | 97.6    | 26.2 | 95.2 | 86.6 | 60.1 | 89.8   | 71.2  | 78.0 |
>
>
> ---
>
> **Reference**
>
> [1] Understanding and improving visual prompting: A label-mapping perspective (CVPR 2023)
>
> [2] AutoVP: An Automated Visual Prompting Framework and Benchmark (ICLR 2024)

---

> > ### Author Response · Authors · 2024-04-13
> > **Re: Review of Paper2367 by Reviewer ExA8**
> >
> > >**Question4:** As mentioned in Sec. 5.2, the results showed that simple augmentation techniques like RandomHorizontalFlip and RandomCrop can already achieve the best results. It is not clear why the proposed method considers other augmentation in the EVP framework.
> >
> > **Answer4:** Sorry for the confusion. As shown in our Ablation Section, we ablate different data augmentation methods. The observation is that simple augmentation is able to improve the generalizability; while strong augmentation may potentially destroy the distribution of target images, thereby disrupting the learning of prompts. We will make this ablation more clear in the next version.
> >
> > >**Question5:** The performance in non-CLIP models is much worse than LP (Table 2). The author could explore not only CNN-based models (i.e., ResNet-50 and Instagram) but also ViT.
> >
> > **Answer5:** Our performance with non-CLIP models is relatively low since we only focus on the input-side prompt while keeping the original classification head fixed. However, with the label-mapping strategy proposed in ILM, our performance can be substantially enhanced, as evidenced by the Table shown in Question3.
> >
> > >**Question6:** The difference in trainable parameter size between EVP-small and EVP-big is not clearly explained. The statement "EVP-big pads pixel patches around the original image" in the caption of Table 5 is confusing; it is not clear whether the term "original image" refers to the dataset's resolution or the input size required by the model. Also, please also add the results of “EVP-small w/o PE.”
> >
> > **Answer6:** Sorry for the confusion. The term "small" refers to our preprocessing step of resizing the image to 164x164, followed by padding the periphery with a prompt, resulting in a final image dimension of 224x224. The designation "big" implies that we resize the image to 224x224 and subsequently pad the surroundings with a prompt to achieve a final image size of 288x288. Within this configuration, our "big" group possesses a higher number of trainable parameters and a higher image resolution. However, in the absence of position embedding, the performance is, paradoxically, inferior to that of the "small" group, suggesting that position embedding is important in securing a strong visual promoting performance. We also added the results of “EVP-small w/o PE.”, and the results are in the Table for Question3.
> >  We will make this part more clear in the next version.
> >
> > >**Requested Changes1:** The improvement in performance resulting from different techniques (augmentation and normalization in Sec. 5.3) has only been validated on CIFAR100. It is unclear whether these techniques would still effectively enhance accuracy on other datasets.
> >
> > **Answer7:** Thanks for your suggestion. Please see the results below on other datasets, which can support our conclusion about the effectiveness of augmentation and normalization. We will add these additional results in the next version.
> >
> >
> >
> > |         | cifar100 | cifar10 | flower102 | food101 | eurosat | sun397  | svhn | pet  | dtd  | resisc | clevr | Avg.  |
> > | ------- | -------- | ------- | ------ | ---- | ------- | ---- | ---- | ---- | ---- | ------ | ----- | ---- |
> > | no aug  | 80.5     | 96.2    | 79.8   | 83.5 | 96.5    | 68.0 | 90.1 | 79.4 | 67.8 | 89.3   | 74.6  | 82.3 |
> > | no norm | 77.5     | 95.7    | 78.5   | 80.9 | 92.9    | 54.3 | 72.3 | 78.7 | 48.2 | 82.6   | 72.1  | 75.8 |
> > | EVP| 81.2     | 96.6    | 82.3   | 84.1 | 97.6    | 71.0 | 90.5 | 90.0 | 68.4 | 89.7   | 75.9  | 84.3 |
> >
> >
> > >**Requested Changes2:** The discussion on different designs of positional embedding could be enhanced. In addition to interpolation and fixed n-th PE, exploring more optimal approaches could provide further insights.
> >
> > **Answer8:** Thanks! We have also explored various strategies. For instance, we have attempted the random generation of position embedding values or treated the position embeddings as independent parameters to be optimized separately from the prompt. However, we failed to identify particularly outstanding strategies. We are happy to briefly discuss these “failures” in the next version.

---

> > > ### Comment · Reviewer_ExA8 · 2024-04-28
> > >
> > > Thank you for the responses and clarifications. Overall, I think the methods are interesting to the community.

---

### Review · Reviewer_dzmD · 2024-03-28

**Summary Of Contributions:**

The paper introduces a visual prompting method, Enhanced Visual Prompting (EVP), for adapting pre-trained models to downstream tasks. The authors propose two main innovations: a non-overlapping strategy for combining prompts with images and applying adversarial example techniques (input diversity and gradient normalization) to visual prompting. The method achieves state-of-the-art performance across multiple datasets, outperforming existing methods like linear probing and VPT in terms of accuracy, robustness to distribution shifts, and corruption, as well as flexibility in handling different data scales.

**Audience:**

Yes

**Claims And Evidence:**

Yes

**Requested Changes:**

See weaknesses

**Strengths And Weaknesses:**

## Strengths
1. The paper presents a significant improvement over the state-of-the-art visual prompting, with empirical evidence showing an average accuracy increase of 5.2% over previous methods.
2. Using adversarial training techniques to improve the optimization and generalization of visual prompts is a creative adaptation from another domain.

## Weaknesses
1. The paper's methodology concatenates three discrete aspects: VPT, augmentation, and normalization. However, it lacks an organic exposition of the relationships between these three points, merely presenting a combination of tricks without elucidating the necessity of integrating the aforementioned three tricks.
2. The lack of comparison with large-scale datasets such as ImageNet could be a limitation.
3. The paper could benefit from a deeper theoretical analysis explaining why the proposed method works better than existing approaches, providing insights beyond empirical results.
4. The ablation studies are helpful, but additional insights into the failure cases or limitations of the method would provide a more balanced view of the EVP's performance.
5. The method's performance on fine-grained datasets like Flowers102 and Food101 is not as strong, suggesting a potential area for further improvement.

---

Overall, the paper makes a certain contribution to the field of visual prompting and the adaptation of pre-trained models, with clear empirical evidence supporting the effectiveness of the proposed methods. However, the paper is merely a stack of common tricks lacking originality. Additionally, it could include more analysis and benchmarking to strengthen the findings and provide a more comprehensive assessment.

---

> ### Author Response · Authors · 2024-04-13
> **Re: Review of Paper2367 by Reviewer dzmD**
>
> We appreciate your considerate review and the chance to respond to the concerns expressed regarding our paper.
>
> > **Question1:** The paper's methodology concatenates three discrete aspects: VPT, augmentation, and normalization. However, it lacks an organic exposition of the relationships between these three points, merely presenting a combination of tricks without elucidating the necessity of integrating the aforementioned three tricks.
>
> >**Question3:** The paper could benefit from a deeper theoretical analysis explaining why the proposed method works better than existing approaches, providing insights beyond empirical results.
>
>
>  **Answer1:**  Thanks for your feedback. We believe these three points are interconnected and they jointly address two key issues of VP. The first is adding a prompt directly to the image like VP may distort the original information of the image, thereby limiting the prompt’s learning potential — therefore we propose to adopt a '’shrink then padding'’ strategy to avoid this issue. The second is that the generalizability of VP is not strong — by noting that the formulation between VP and adversarial attack is similar, we re-introduce two “old tricks” in transferable adversarial attack (i.e.,  gradient normalization and data augmentation) to enhance VP.
>
> We will make these interconnections more clear in the next version.
>
>
> >**Question2:** The lack of comparison with large-scale datasets such as ImageNet could be a limitation.
>
> **Answer2:** Thanks for your suggestion. We note that our EVP can still outperform other visual prompting methods when testing on ImageNet. For example, VP achieves an accuracy of 58.6%, VPT achieves an accuracy of 59.8%, and our EVP achieves the highest accuracy of 60.1%. However, we note the performance here is still relatively low, suggesting that generalizing visual prompting algorithms to the large-scale ImageNet is still very challenging (which we leave as future work).
>
> > **Question4:**  The ablation studies are helpful, but additional insights into the failure cases or limitations of the method would provide a more balanced view of the EVP's performance.
>
> > **Question5:** The method's performance on fine-grained datasets like Flowers102 and Food101 is not as strong, suggesting a potential area for further improvement.
>
> **Answer3:** Thanks for your suggestion. One major failure case of our EVP is its limited performance when applied to non-CLIP models. This issue primarily arises from its optimization focusing exclusively on input pixels, resulting in suboptimal outcomes due to the uncalibrated final layer (i.e., classification head) of non-CLIP models.
> One potential solution is to employ the label mapping strategy proposed by ILM[1]. We conducted experiments under the setting with ViT-Base16 as the backbone, and the results are presented in the table below. We can observe that the integration of ILM results in a substantial performance improvement of both VP and EVP. Moreover, irrespective of the use of ILM, EVP consistently outperforms VP.
> We will add these discussions in the next version.
>
>
>
> |            | cifar100 | cifar10 | flower102 | food101 | eurosat | sun397  | svhn | pet  | dtd  | resisc | clevr | Avg.  |
> | ---------- | -------- | ------- | ------ | ---- | ------- | ---- | ---- | ---- | ---- | ------ | ----- | ---- |
> | VP  | 44.6     | 94.9    | 15.3   | 36.5 | 95.3    | 2.2  | 16.2 | 8.8  | 16.8 | 64.8   | 35.7  | 39.2 |
> | EVP | 73.8     | 97.7    | 78.6   | 62.7 | 97.6    | 5.7  | 19.6 | 8.7  | 57.4 | 89.5   | 50.2  | 58.3 |
> | ILM+VP     | 79.3     | 97.0    | 36.9   | 42.6 | 96.0    | 17.6 | 89.7 | 79.5 | 36.2 | 69.6   | 64.0  | 64.4 |
> | ILM+EVP    | 86.3     | 97.6    | 77.5   | 70.3 | 97.6    | 26.2 | 95.2 | 86.6 | 60.1 | 89.8   | 71.2  | 78.0 |
>
>
>
>
> ---
>
> **Reference**
>
> [1] Understanding and improving visual prompting: A label-mapping perspective (CVPR 2023)

---

> > ### Comment · Reviewer_dzmD · 2024-04-13
> >
> > Thank you for your reply. It effectively addressed some of my concerns.

---

### Review · Reviewer_WuQV · 2024-03-29

**Summary Of Contributions:**

This paper presents an improved visual prompt method. The main idea is to first resize the image, then augment the image, and finally train the prompt with gradient normalization. This method is implemented on many tasks and datasets and shows advantages over the classical visual prompt method and linear probing to some degree.

**Audience:**

No

**Claims And Evidence:**

Yes

**Requested Changes:**

1, Can you provide a theoretical or intuitive understanding of how the used tricks help the visual prompt method? Are there any new insights into these tricks for the visual prompt method?

2. Can you show the comparison with the Linear Probing method and the Visual Prompt method, in terms of the memory cost, time cost, and throughput?

**Strengths And Weaknesses:**

Strengths:

1. The paper is easy to follow and well-written.
2. The experiments are complete and solid.

Weaknesses:

1. Some motivations and explanations of the proposed method are needed. For example, why do you use gradient normalization? Are there any findings related to the tricks used, like augmentation and gradient normalization, that are special to visual prompts?
2. The novelty is a limitation since the tricks are all old ones. However, novelty is not a focus of this venue.

---

> ### Author Response · Authors · 2024-04-13
> **Re: Review of Paper2367 by Reviewer WuQV**
>
> Thanks for your feedback. We appreciate the effort you have put into reviewing our work and the insightful comments you have shared. Your constructive criticism has undoubtedly contributed to the improvement of our research.
>
> >**Question1**: Some motivations and explanations of the proposed method are needed. For example, why do you use gradient normalization? Are there any findings related to the tricks used, like augmentation and gradient normalization, that are special to visual prompts?
>
> > **Requested Changes1:** Can you provide a theoretical or intuitive understanding of how the used tricks help the visual prompt method? Are there any new insights into these tricks for the visual prompt method?
>
> **Answer1:** Thanks for raising this concern. To clarify, our motivation is twofold. Initially, we observed that the integration of the prompt in the VP framework disrupts the information contained within the original image, thus leading us to adopt a "shrink then padding" strategy. Secondly, the formulation of prompts has notable parallels with that of adversarial attacks. Therefore, we employed gradient normalization and data augmentation, which are commonly used techniques from transferable adversarial attacks and observed that these techniques empirically can enhance the performance of the visual prompting task. We will make these points more clear in the next version.
>
>
> > **Requested Changes2:** Can you show the comparison with the Linear Probing method and the Visual Prompt method, in terms of the memory cost, time cost, and throughput?
>
> **Answer2:** Thanks for this suggestion. Please see these metrics on our CIFAR100 experiments with the CLIP model below, where we listed the training time, throughput, and the number of parameters of vp, vpt, evp, and linear probe, respectively.
>
> |Adaptation|Parameters (Millions)|Training time (mins)|Throughout (images/second)|
> |---|---|---|---|
> |VP|0.070|358|780|
> |VPT|0.064|1050|651|
> |EVP|0.062|409|961|
> |Linear probing|0.051|356|1130|

---

> > ### Comment · Reviewer_WuQV · 2024-04-19
> >
> > Thank you for the clarification. This makes the paper easier to understand.

---

### Review · Reviewer_QkpB · 2024-03-29

**Summary Of Contributions:**

The authors proposed a new visual prompting method called Enhanced Visual Prompting (EVP), that is motivated from the Visual Prompting and combined with different training strategy like diversified input and gradient normalization. The authors provide thorough experiments to demonstrate the effectiveness of the proposed method.

**Audience:**

Yes

**Broader Impact Concerns:**

N/A.

**Claims And Evidence:**

Yes

**Requested Changes:**

I would like to see a more systematic study on the relevant training strategies, as I mentioned in the weakness part.

**Strengths And Weaknesses:**

###  Strengths:
* The idea is easy to follow.
* The perspective on the connection between adversarial examples and visual prompting is interesting.
* Ablation results are interesting, for example the importance of positional embedding.

### Weaknesses:
* The source of the improvement is not clear. For example, I would wanna know if the diversified input and the gradient normalization can be applied to VP and obtain some improvement.
* Although there are some interesting observations, the discussions are not thorough. For example, would it be much better to introduce some prompts in the middle of the image, if the positional embedding is important?

---

> ### Author Response · Authors · 2024-04-13
> **Re: Review of Paper2367 by Reviewer QkpB**
>
> Thank you for the encouraging review. In the following we hope to address all your concerns one by one.
>
> >  **Question 1:** The source of the improvement is not clear. For example, I would wanna know if the diversified input and the gradient normalization can be applied to VP and obtain some improvement.
>
> **Answer1:** Thanks for raising this concern. Following your suggestion, we have conducted experiments to furnish a more comprehenvise substantiation of our approach. By incorporating our augmentation and normalization strategies atop the VP framework, we have achieved strong improvements in comparison to the baseline VP performance. For example, as shown in the Table below, the average accuracy is boosted from 76.5% to 79.4% (+2.9%).
>
>
> | Adaption| cifar100| cifar10| flower102| food101| eurosat| sun397| dmlab| svhn| pets| dtd| resisc| clevr| Avg. |
> | --- | --- | --- | --- | --- | --- | --- | --- | --- | ---- | ---- | ---- |---- | ---- |
> | VP | 75.3 | 94.2 | 62.0 | 83.2 | 95.6 | 68.4 | 41.9 | 88.4 | 86.5 | 57.1 | 84.1 | 81.4 |76.5 |
> | VP+Aug&norm | 78.4 | 95.0 | 74.7 | 80.0 | 96.5 | 68.0 | 43.7 | 88.5 | 90.0 | 64.8 | 88.8 | 84.7 | 79.4 |
>
>
> > **Question2:** Although there are some interesting observations, the discussions are not thorough. For example, would it be much better to introduce some prompts in the middle of the image, if the positional embedding is important?
>
> **Answer2:** Thanks for your suggestion. Firstly, we would like to clarify that one of our key observations is that introducing prompts within the image like VP may disrupt the original image information, thereby limiting the prompt’s learning potential. Therefore, we instead propose to shrink the input images into a smaller size and then pad the prompt around it in order to preserve the original image information.
> In order to substantiate this design choice, we conducted experiments delineated in Section 5.1. Specifically, we maintained a fixed prompt size while varying the resolution of the original image. Given that the final output resolution is 224*224, an increase in the size of the original image results in higher overlap with the prompt. Our findings indicate that an elevated degree of overlap correlates with diminished performance, thereby supporting the importance of preserving the original image information within our strategy.
> We will make these observations/discussions more clear in the next version.

---

### Decision · Action_Editor_diMp · 2024-04-28

**Recommendation:** Accept with minor revision

**Comment:**

The recommendation is based on the reviewers' comments, the action editor's evaluation, and the authors’ response.

This paper proposed methods to improve visual prompting (VP) based on enhancing input diversity and gradient normalization, inspired by prior studies on adversarial examples. The results show improved empirical performance (on average and by large) over standard VP. All reviewers agreed that the paper provides solid results and empirical evidence while sharing a common comment on lacking sufficient technical novelty (given that these techniques are known methods in alternative topics). However, the contributions are modest and meet the TMLR Acceptance Criterion (https://jmlr.org/tmlr/acceptance-criteria.html).

Some related and concurrent works on VP were not discussed in the paper.
- Visual prompting was firstly studied in the context of model reprogramming https://arxiv.org/abs/2007.08714
- A recent (or concurrent) work also studied methods to improve VP, especially for the effect of the prompt size. https://arxiv.org/abs/2310.08381


Overall, I recommend acceptance of this submission with minor revision. I expect the authors to expand the discussion on relevant works and include the new results and suggested changes during the rebuttal phase to the final version.

**Audience:**

Of broad interest to computer vision and parameter-efficient fine-tuning of vision foundation models.

**Claims And Evidence:**

The claims of improved performance over the standard visual prompting are supported by the empirical results.